# Quantitative Evaluation System of Wrist Motor Function for Stroke Patients Based on Force Feedback

**DOI:** 10.3390/s22093368

**Published:** 2022-04-28

**Authors:** Kangjia Ding, Bochao Zhang, Zongquan Ling, Jing Chen, Liquan Guo, Daxi Xiong, Jiping Wang

**Affiliations:** 1School of Biomedical Engineering (Suzhou), Division of Life Sciences and Medicine, University of Science and Technology of China, Suzhou 215163, China; dingkj@mail.ustc.edu.cn (K.D.); zbc2020@mail.ustc.edu.cn (B.Z.); lzq597@mail.ustc.edu.cn (Z.L.); sle1008@mail.ustc.edu.cn (J.C.); guolq@sibet.ac.cn (L.G.); xiongdx@sibet.ac.cn (D.X.); 2Suzhou Institute of Biomedical Engineering and Technology, Chinese Academy of Sciences, Suzhou 215163, China

**Keywords:** stoke, quantitative evaluation, wrist motor, force feedback, machine learning

## Abstract

Motor function evaluation is a significant part of post-stroke rehabilitation protocols, and the evaluation of wrist motor function helps provide patients with individualized rehabilitation training programs. However, traditional assessment is coarsely graded, lacks quantitative analysis, and relies heavily on clinical experience. In order to objectively quantify wrist motor dysfunction in stroke patients, a novel quantitative evaluation system based on force feedback and machine learning algorithm was proposed. Sensors embedded in the force-feedback robot record the kinematic and movement data of the subject, and the rehabilitation doctor used an evaluation scale to score the wrist function of the subject. The quantitative evaluation models of wrist motion function based on random forest (RF), support vector machine regression (SVR), k-nearest neighbor (KNN), and back propagation neural network (BPNN) were established, respectively. To verify the effectiveness of the proposed quantitative evaluation system, 25 stroke patients and 10 healthy volunteers were recruited in this study. Experimental results show that the evaluation accuracy of the four models is all above 88%. The accuracy of BPNN model is 94.26%, and the Pearson correlation coefficient between model prediction and clinician scores is 0.964, indicating that the BPNN model can accurately evaluate the wrist motor function for stroke patients. In addition, there was a significant correlation between the prediction score of the quantitative assessment system and the physician scale score (*p* < 0.05). The proposed system enables quantitative and refined assessment of wrist motor function in stroke patients and has the feasibility of helping rehabilitation physicians in evaluating patients’ motor function clinically.

## 1. Introduction

A stroke is a neurological defect caused by acute focal injury of the central nervous system caused by vascular causes, including cerebral infarction, cerebral hemorrhage, etc. It is one of the leading causes of disability and death worldwide [1]. Globally, the risks of stroke due to an aging population and the accumulation of risk factors continue to rise [2]. Most surviving stroke patients have varying degrees of dysfunction, such as motor dysfunction, sensory dysfunction, and cognitive and language impairments [3]. Upper limb motor and wrist dysfunctions caused by strokes seriously affect the normal working life and life of stroke patients. The improvement and rehabilitation of upper limb motor function are core elements of rehabilitation for stroke patients, which can significantly improve the rehabilitation effects of patients and reduce disability [4]. Currently, the rehabilitation training of upper limb functions after experiencing a stroke and evaluation methods have been proved by a large number of studies and experiments [5], but the distal upper-limb wrist motor function has received less attention in the field of rehabilitation training and evaluation. Effective intervention and rehabilitation research of wrist motor function in patients with stroke is of great significance to improve their quality of life and to maximize the recovery of the overall function of the upper limbs after strokes.

Currently, rehabilitation physicians’ one-to-one scale evaluation method is often used to evaluate upper limb and wrist function in clinical practice, with shortcomings such as being time consuming, strong subjectivity, and coarse grading [6]. With a series of rehabilitation robots being developed and applied to actual clinical rehabilitation, further development in rehabilitation and evaluation techniques of upper limb and wrist motor functions in stroke patients is needed [7]. Studies have shown that haptic or force feedback provided by assisted rehabilitation robots can effectively improve the rehabilitation efficiency of stroke patients [8,9,10]. Force feedback has an important role in the development of therapeutic focused on ensuring synchronization between the graphical environment, the location of the target, and the sensory system, but the magnitude and direction of forces emitted by the device must be controlled to facilitate direct rehabilitation and will not cause collateral damage to the limbs during exercise [11]. Xu et al. [6] recruited 40 stroke patients to explore the effects of rehabilitation robot training on upper limb motor function and daily activities of stroke patients and demonstrated that rehabilitation robot training could significantly contribute to upper limb motor function and daily activities in subacute stroke patients. Andaluz et al. [8] used Novint Falcon, a haptic robot that can provide force feedback, combined with Oculus Rift and Leap motion sensors to develop an intelligent system for upper limb fine motor function rehabilitation and validated the effectiveness of the human–computer interaction system. Cappa et al. [12] demonstrated that force feedback can modulate the smoothness, accuracy, and duration of subjects’ movements. Mauricio Tamayo et al. [13] used force feedback based on the Novint Falcon along with a virtual environment to provide effective rehabilitation for carpal tunnel syndrome. These related studies provide a strong premise and evidence for this paper, indicating that force feedback can play an excellent auxiliary role in the evaluation of wrist motor function for stroke. At the same time, it is shown that the haptic force feedback device Novint Falcon can provide reliable force feedback for rehabilitation training and assist patients in the rehabilitation and treatment of motor function.

The evaluation of motor function is a significant part of the rehabilitation of stroke patients. Researchers and therapists have developed functional evaluation scales such as the Brunnstrom scale and Fugl-Meyer scale [14,15]. As smart devices such as rehabilitation robots have improved rehabilitation training methods, evaluating a patient’s motor function needs to be further improved. Studies have shown that robot-assisted rehabilitation technology with integrated machine learning algorithms has great promise for the rehabilitation and evaluation of stroke patients [7]. Yang et al. [16] designed an IoT-based stroke rehabilitation system based on an intelligent wearable armband, machine learning algorithms, and 3D-printed robotic dexterous hands to achieve an accurate recognition of stroke patients’ hand gestures and to achieve the rehabilitation evaluation of stroke patients’ motor function. Chae et al. [17] developed a home-based rehabilitation system that identifies and records the type and frequency of rehabilitation training performed by users using a smartwatch and smartphone app equipped with a convolutional neural network, and experimental results demonstrated that this system could facilitate stroke patients’ participation in-home rehabilitation training and can provide patients with scores related to motor function improvement. Zollo et al. [18] performed multimodal analysis of stroke patients undergoing upper limb rehabilitation training by using wearable sensors and rehabilitation robotics, thus quantifying the biomechanical and motor characteristics of patients during rehabilitation and enabling the quantitative evaluation of upper limb motor function. Otten et al. [19] proposed a framework for automating upper-limb motor assessments that use low-cost sensors to collect movement data and machine learning algorithms to process the data to obtain a score of the patient’s upper extremity function.

Although many researchers have conducted studies on the quantitative evaluation of motor function in stroke patients, the results vary due to the different motor parameters selected in the process of evaluation, which poses a greater challenge to motor function quantitative evaluation studies. The intelligent rehabilitation robot equipment used by some researchers is more complex and expensive, making it difficult to achieve portability and to extend it to clinical applications as well as remote rehabilitation and evaluation. In addition, there are few studies on intelligent rehabilitation training and quantitative evaluation of the wrist at the end of the upper limb, which has significant implications for improving the quality of life and daily living standard of stroke patients.

Regarding the problems raised above, a quantitative wrist motor function evaluation system based on force feedback and machine learning algorithms was proposed in this paper. The low-cost haptic force feedback robot Novint Falcon was selected for wrist function evaluation, and a virtual reality interactive system was designed based on force feedback and visual feedback. The subjects manipulate the control handle to complete the trajectory tracking task and record the data, such as the trajectory coordinates of the handle and the time of executing the task. By conducting data processing and analysis, indicators that can characterize wrist motion, such as the number of peak velocity points, average acceleration, and track coincidence, were selected. In addition, the article proposed a quantitative evaluation model based on machine learning algorithms to achieve a quantitative evaluation of the wrist motor function of stroke patients. The haptic force feedback motion mode and virtual reality system proposed in this paper enable stroke patients to have greater willingness and interest in participating in the evaluation of wrist motor function. The use of robot-assisted rehabilitation technology and machine learning algorithms can simplify and intellectualize motor function evaluations, which is more conducive for rehabilitation clinicians to provide accurate and targeted rehabilitation training and guidance for patients.

## 2. Materials and Methods

### 2.1. System Framework

The system block diagram of the quantitative wrist motor function evaluation system proposed in the article is shown in Figure 1, and it is divided into three main parts: data acquisition, machine learning model training, and quantitative assessment.

In the data acquisition phase, stroke patients manipulate Novint Falcon’s end effector to complete the task of wrist motor function evaluation. The commercial haptic Novint Falcon device provides the patient with force feedback of various magnitudes, and the PC-based control software provides patients with visual feedback of real-time trajectories. As the patient performs the task, Novint Falcon’s built-in sensors record trajectory coordinate changes of the end-effector as well as the magnitude of the force feedback and the time spent performing the task in real-time; it then transmits the collected data to the PC-based control software. At the same time, the wrist motor function of stroke patients is scored and recorded by an experienced rehabilitation physician using a clinical assessment scale.

In the machine learning algorithm models training phase, all machine learning algorithms displayed in the figure were used to establish evaluation models and to conduct model training. The data collected by the sensors were first subjected to data preprocessing; the features related to the wrist motion function were extracted from the motor data to obtain a feature dataset. We selected filtering and normalization for data processing and compared the effects of using the two methods separately and using the two methods combined for data processing. The evaluation model obtained higher accuracy and performance using only normalization. Therefore, we selected normalization for the method of data processing. Four machine learning algorithms, including random forest regression, support vector machine regression, K-nearest neighbor regression, and Back Propagation neural network, were selected to build the evaluation and prediction model for wrist motor function. The score of rehabilitation physicians was used as the label for models training, and the motion feature dataset obtained by processing was used as the model’s input for training the regression prediction models. The models with high regression prediction performance were trained by adjusting the parameters of each model.

In the quantitative assessment phase, the participating subjects completed the evaluation task, and their wrist motion data were collected by sensors and transmitted to the computer, and then the model prediction score of the patient’s wrist motion function was obtained by using the trained model after data pre-processing, which enables the quantitative evaluation of the wrist motion function of stroke patients.

### 2.2. Experimental Device

The commercial haptic force feedback robot Novint Falcon (Novint Technologies, Inc., Albuquerque, NM, USA) was selected for the wrist motor function evaluation trial, as shown in Figure 2a. It is a low-cost 3-DOF haptic force feedback device in a form similar to the Delta robot configuration, enabling it to move in three dimensions. The Novint Falcon can generate force feedback in the X-Y-Z plane of the reference coordinate system with magnitudes ranging from 0 to 8.8 N. The design of the Falcon also includes a detachable end-effector, and the working space of the end effector is approximately 10 × 10 × 10 cm^3^. The Novint Falcon communicates with the control computer using a USB interface that uses a 1 kHz frequency for data sampling [20].

The Novint Falcon is equipped with a detachable standard spherical handle, but it is not suitable for stroke patients with wrist motor dysfunction, a cylindrical handle for stroke patients with easy grip was designed using a 3D printing method [21,22]. Figure 2b is the design rendering of the modified handle, and Figure 2c is the rendering of the modified handle installed on the Novint Falcon. Barak et al. [23] showed that using a 3D-printed modified attachment on the Novint Falcon aided wrist motion without increasing the motion compensation of the shoulder and elbow of stroke patients and improved participant satisfaction and willingness to repeat the exercise.

The experimental device in the quantitative evaluation system consists of the haptic force feedback robot and the upper computer control system. In this paper, LabVIEW (National Instruments, Austin, TX, USA) was used to develop the upper computer control system of Novint Falcon. As shown in Figure 3, the data acquisition interface of Novint Falcon’s upper computer control system includes four parts, which are used to help the subjects to complete the track following task and adjust the position of Novint Falcon’s end effector. The red rectangular box in Figure 3a shows the real-time trajectory in the X-Y plane compared with the graphical curve given by the experimental task; the yellow box in Figure 3b shows the coordinate change of the Novint Falcon’s modified handle in the Z-axis of the spatial coordinate system, which is used to assist the subject to better manipulate the handle to complete the experimental task; the blue box in Figure 3c shows the saved motion trajectory of the subject in the experiment; the green box in Figure 3d shows the operation buttons and data output part of the upper computer control system, including the data saving button, the stop experiment button and the force feedback magnitude, the position coordinates of the handle in the X-Y coordinate system, and the time of the task execution.

### 2.3. Experimental Protocol

To test and verify the quantitative wrist motor function evaluation system proposed in this paper, a clinical experimental protocol was designed. The subjects were recruited to participate in the wrist motor function evaluation experiment, and the real scene of the subjects who participated in the experiment is shown in Figure 4a, which clearly shows the location of the haptic force feedback robot and the upper control system computer in the quantitative evaluation experiment. The Novint Falcon, a haptic force feedback robot with a modified handle, was placed on a table, and the subjects sat in front of the Novint Falcon, and a computer was placed on the table for the display of the experimental interface and visual feedback to the subjects. Figure 4b,c show that a female and a male subject used different hands to complete the track following task, respectively, in the wrist motor function assessment experiment.

The clinical evaluation scale in the experiment consists of the wrist function part of the Fugl-Meyer scale and the wrist function part of the Nottingham Sensory Assessment Scale, which is commonly used in clinical practice according to the experienced rehabilitation physicians [14,24]. The assessment scale contains 14 tasks, and each task has a full score of 2. A score of 0 indicates that the wrist movement cannot complete the task action, a score of 1 indicates that the wrist movement can complete the task action partially, and a score of 2 indicates that the wrist movement can complete the action normally. Before the start of the experiment, the experienced rehabilitation physicians were invited to evaluate the subjects’ wrist motor function using the clinical evaluation scale, and the subject was given instructions by the rehabilitation doctor on experimental tasks and is familiar with the experimental equipment and procedures. The rehabilitation scale score, which scored by the physician, was recorded and used as a label for model training.

Barak et al. [23] demonstrated that using a 3D-printed modified attachment to replace the original end-effector of the Novint Falcon assisted patients’ wrist movements without increasing the motion compensation of the shoulder and elbow of stroke patients. Morevoer, under the training guidance of the rehabilitation doctor, subjects were asked to minimize shoulder motion and compensations while manipulating the modified handle to complete the experimental task and to minimize the interference of compensation on the evaluation of the patient’s wrist motor function. Cappa et al. [12] used a predefined trajectory-following task in their study to quantify subjects’ dexterity in planning and to generate a multi-joint visuomotor task. Elena et al. [25] asked subjects to control the movement of a sphere along a desired path in the virtual environment and obtained the subjects’ performance in motion by analyzing kinematic data from the trajectory-following task. Thus, the trajectory-following task is suitable for describing the motor function performance of subjects in related studies.

At the beginning of the experiment, the subjects were asked to hold the Novint Falcon’s end modification handle, follow the graphical trajectory displayed on the screen in a clockwise motion, and make the end-effector’s motion trajectory coincide with the graphical trajectory as much as possible. During the experiment, the Novint Falcon’s built-in sensors recorded the coordinate change of the end-effector handle in the X-Y plane, as well as the completion time of each trajectory-following task, and transmitted the data to the upper computer control program, which displayed the given graphical curve in red and the real-time trajectory of the subject’s manipulation of the handle in blue, as shown in Figure 4d.

To comprehensively evaluate wrist motion function, a force feedback-based trajectory following task was designed in this study. The experiment includes three force feedback modes: (1) no force feedback 0 N; (2) the smaller force feedback 1.5 N; (3) the larger force feedback 5 N. The curves’ shape used for the trajectory following included square, triangle, and circle. The subjects were required to perform three tasks with force feedback at 0 N, 1.5 N, and 5 N, respectively. The subjects need to perform the trajectory following-tasks of the three task graphical curves under three force feedback modes, respectively (that is, each subject needs to complete the trajectory-following task nine times and record experimental data). The wrist movements mainly involved in the trajectory following task were wrist flexion and extension, wrist dorsiflexion, and wrist circular motion. Figure 4d shows the comparison between the real-time trajectory of the subject and the given task graphical curve during the experiment.

### 2.4. Participants

The experiment was conducted in Suzhou Xiangcheng People’s Hospital. This study was approved by the Ethics Committee of Suzhou Xiangcheng People’s Hospital (Ethics Committee approval number 2018 (006)), and each subject signed a written informed consent prior to inclusion in the study. Twenty-five patients (sixteen males; nine females) with impaired wrist motor function after stroke were recruited from Suzhou Xiangcheng People’s Hospital, while ten healthy volunteers (six males; four females) were recruited to participate in the experiment as a control group. The inclusion criteria of patient selection are as follows: (1) patients between the ages of 18 and 80 years; (2) patients with a first ischemic or hemorrhagic stroke; (3) patients with the upper limb Brunnstrom stage II or above; (4) patients without major post-stroke complications and without severe cognitive deficits; (5) patients who are able to participate rehabilitation treatment and sign informed consent. The exclusion criteria of patient selection are as follows: (1) patients with recurrent stroke; (2) patients with aphasia, cognitive impairment, and psychiatric symptoms, etc.; (3) patients with hemiplegia or upper limb impairment that prevented normal participation in the experimental procedure. The inclusion criteria for volunteers in the control group are healthy, with no history of neurological or musculoskeletal-related diseases. Considering the differences between the motor behaviors of the dominant and non-dominant hand, ten control volunteers recruited to participate in the experiment were asked to use the dominant hand and the non-dominant hand, respectively. The experienced rehabilitation physicians were invited to evaluate the subjects’ wrist motor function using the clinical evaluation scale. The scale score of the subject scored by the clinical doctor and the patient’s demographic information is shown in Table 1.

### 2.5. Data Preprocess

#### 2.5.1. Feature Extraction

Research on the motor performance of stroke patients has identified several indicators to quantify the patient’s related motor function, including smoothness and coordination of movement. Cappa et al. [12] selected velocity metric, number of sub-movements, trajectory deviation, and normalized path length in characterizing motor performance. Gutiérrez et al. [22] used average velocity and peak velocity, number of peak points on the trajectory, and duration of the task as characteristic parameters to evaluate the performance of patients’ upper limb movements. Emilia et al. [24] used movement duration, the ratio of the length of the subject’s actual path to the length of the target path, trajectory deviation, velocity metric, and normalized acceleration to characterize subjects’ motor performance during the experiment. Elena F. Gambaro et al. [25] selected the area between the actual path and the desired path as an index to characterize motor function in the study of the effectiveness of vibrotactile feedback.

Referring to the above-mentioned related studies on the characterization of motor performance and the professional advice of rehabilitation physicians, this paper selected seven motor feature parameters, such as the number of peak velocity points in the evaluation of wrist motor function for stroke patients. The scripts for data processing were developed using MATLAB (MathWorks, Natick, MA, USA), which has been uploaded to Github (https://github.com/DKJ00/Quantitative-evaluation-of-wrist-motor-function, accessed on 24 March 2022). The symbols and definitions of the specific feature parameters are shown in Table 2.

The average velocity is calculated as follows:(1)$$V=\frac{1}{N}\left(\sum_{i=1}^{N}V_{i}\right)$$
where N is the number of sampling points on the real-time trajectory, and $V_{i}$ is the instantaneous velocity at the sampling point.

The average acceleration is calculated as follows:(2)$$A=\frac{1}{N}\left(\sum_{i=1}^{N}A_{i}\right)$$
where N is the number of sampling points on the real-time trajectory, and $A_{i}$ is the instantaneous acceleration at the sampling point.

The average trajectory deviation is calculated as follows:(3)$$D=\frac{1}{N}\left(\sum_{i=1}^{N}\left|\Delta D_{i}\right|\right)$$
where N is the number of sampling points on the real-time trajectory, and $\Delta D_{i}$ is the deviation between the point on the real-time trajectory and the point corresponding to the task curve.

The trajectory coincidence is calculated as follows:(4)$$C=\frac{L_{c}}{L_{r}}\times 100\%$$
where $L_{c}$ is the coincidence length between the real-time trajectory and the given task curve, and $L_{r}$ is the length of the real-time trajectory.

The intersecting area of the trajectory is calculated as follows:(5)$$S=\sum_{i=1}^{N_{s}}\Delta S_{i}$$
where $N_{s}$ is the number of graphics formed by the intersection between the real-time trajectory and the given task curve, and $\Delta S_{i}$ is the area of each intersection graphic.

#### 2.5.2. Normalization

Among the selected motor feature indicators, seven features characterizing the motor function have different dimensional units, which will impact the evaluation model’s performance. In order to better process and analyze the data using the quantitative evaluation model established based on the machine learning algorithm, we used normalization for the seven feature variables to eliminate the dimensional influence of features, and data standardization is needed to solve comparability between data indicators [26]. After data were processed by data standardization, the indicators were in the same order of magnitude, which was suitable for comprehensive evaluation and comparison, and also facilitated the subsequent establishment of quantitative evaluation models based on machine learning algorithms. The normalized formula is described as follows:(6)$$x_{i}=\frac{x-\bar{x}}{\sigma \left(x\right)}$$
where x denotes data features, $\bar{x}$ is the mean of the data features, σ(x) is the standard deviation of the data features, and $x_{i}$ is the data features obtained by normalization.

### 2.6. Evaluation Models Establishment

The establishment of evaluation models requires the selection of appropriate machine learning algorithms for regression prediction. By establishing the relationship between motor features and the scale score, the evaluation model was trained to achieve a quantitative evaluation of the wrist movement function in stroke patients. Wang et al. [27] selected three machine learning algorithms as support vector machine, BP neural network, and random forest as the classification evaluation model of multimodal fusion in the study of upper limb motor function of post-stroke hemiplegia patients and verified the accuracy of the classification and evaluation results of the multimodal fusion framework. Lee et al. [28] selected neural networks based on MLP (multilayer perceptron) and RBFN (radial basis function network), support vector machine, and decision tree as the machine learning algorithms in their study to establish a classification evaluation model. The experimental results showed that MLP neural networks could achieve the highest classification accuracy.

Twenty-five eligible stroke patients and ten healthy volunteers were recruited as the experimental group and the control group in this paper. Each subject needs to complete the trajectory following task nine times, and the volunteers need to complete it with their left and right hand to eliminate the influence of the dominant hand, which means that the sample size of the experimental data set is 405 in total. The data set of 35 subjects was randomly divided into a training set and a test set. In this study, four supervised machine learning algorithms, including random forest, support vector machine regression, K nearest neighbor, and Back Propagation neural network, were selected as regression algorithms for establishing quantitative evaluation models. The reason for choosing the above four algorithms is that the generalization ability and nonlinear mapping ability are good [29], and the sample size of the dataset in this study is small.

#### 2.6.1. RF Model

Random forest (RF) is easy to interpret and can rapidly learn; these abilities made it popular in the tele-rehabilitation domain and especially in multi-class activity recognition problems [30]. The principle of feature evaluation in the random forest is to evaluate the contribution value of each feature on each tree in the random forest and then average the contribution values and compare the contribution values between different features; the calculation formula of feature importance is described as follows:(7)$$VIM_{jm}^{\left(Gini\right)}=GI_{m}-GI_{l}-GI_{r}$$
where j is a characteristic of the random forest model, m, l, and r are the nodes of the random forest, and Gini is the index of the random forest.

#### 2.6.2. SVR Model

Support Vector Machine Regression (SVR) is mainly used for clinical assessments where participants are given a clinical score [27,31]. SVR has different kernel types that allow it to deal with linear and non-linear problems, and it has a good generalization ability for sequential data structures and datasets that are not too large. The following is defined:(8)$$f\left(x\right)=\omega^{T}x+b$$
where $\omega \in R^{n}$ is a high dimensional weight vector, and the offset b∈R is a scalar. Note that the sample data set is X=(x,y). By applying the Karush–Kuhn–Tucker theorem, the Lagrange function is defined to solve the SVR problem [32]. The regression score of the SVR model y can be calculated by the following equation:(9)$$y=f\left(x\right)=\sum_{i}^{m}\left({\hat{\alpha }}_{i}-\alpha_{i}\right)K\left(x,x_{i}\right)+b$$
where ${\hat{\alpha }}_{i}\;and\;\alpha_{i}$ are Lagrange multipliers, and $K\left(x,x_{i}\right)$ is a kernel function.

#### 2.6.3. KNN Model

K-Nearest Neighbor (KNN) is based on distance metrics and is widely used in real-time applications as it is free from the underlying assumptions about the distribution of the dataset. The different values of ‘K’ can contribute to different results for the same problem, which makes it an additional hyperparameter for finding the highest performing model, especially in activity recognition [30]. KNN regression needs to find the ‘K’ nearest neighbors closest to the test points in the model establishment. Traverse each point in the dataset and calculate the Euclidean distance L between it and sample X in the test set. The calculated formulation is as follows:(10)$$L=\sqrt{{\left(F_{i}-F\right)}^{2}+{\left(V_{i}-V\right)}^{2}+{\left(\alpha_{i}-\alpha \right)}^{2}+{\left(T_{i}-T\right)}^{2}}$$
where the dataset is X=(F,V,α,T,y), and y is the regression score of KNN model.

#### 2.6.4. BPNN Model

Back Propagation Neural Network (BPNN) is a set of Machine Learning algorithms inspired by the brain’s neurons. Multilayer perceptron (MLP) was the most common method of the interconnection of layers of nodes. It does not require feature engineering and, thus, necessitates less domain expertise. It can achieve a good result for activity recognition and predictability regression [30]. The BPNN model built in this paper has one input layer, three hidden layers, and one output layer. By conducting preliminary data testing and model parameters adjustment, we obtained the optimal number of neurons for the BPNN model. The input layer has seven neurons; the three hidden layers have 56, 28, and 7 neurons, respectively; the output layer O has one neuron. In model training, the stochastic gradient descent is used as the optimization method, and the RELU function is used as the activation function.

### 2.7. Model Evaluation Index

Zhang M et al. [31] proposed an accuracy index to indicate the prediction accuracy of quantitative evaluation models based on machine learning algorithms in the study of the quantitative evaluation of upper limb motor function. On the other hand, rehabilitation therapists believed that the absolute error between the prediction score of the quantitative evaluation model and the doctor’s scale score should be within 3 points. Therefore, in this paper, the prediction score of the quantitative evaluation model was considered correct when the absolute error between the prediction score of the model and the doctor’s scale score was less than or equal to 3. To compare the performance of the wrist motor function quantitative evaluation model established by the four machine learning algorithms proposed in the study, the performance of the models in quantitative assessment was evaluated using accuracy, mean absolute error, mean square error, and determination coefficient [33].

Accuracy is defined as the percentage of the number of samples with absolute error less than 3 of the total number of samples in the test set, and the formula is as follows.
(11)$$Accurary=\frac{N_{s}}{N_{t}}\times 100\%$$

The mean absolute error is calculated as follows.
(12)$$MAE=\frac{1}{n}\sum_{i=1}^{n}\left|\left(y_{i}-\hat{y_{i}}\right)\right|$$

The mean square error is calculated as follows.
(13)$$MSE=\frac{1}{n}\sum_{i=1}^{n}{\left(y_{i}-\hat{y_{i}}\right)}^{2}$$

The determination coefficient is calculated as follows:(14)$$R^{2}=1-\frac{\sum_{i=1}^{n}{\left(y_{i}-\hat{y_{i}}\right)}^{2}}{\sum_{i=1}^{n}{\left(y_{i}-\bar{y_{i}}\right)}^{2}}$$
where $N_{s}$ is the number of samples with absolute error less than 3, $N_{t}$ is the total number of samples in the test set, $y_{i}$ is the doctor’s scale score, $\bar{y_{i}}$ is the mean of $y_{i}$, and $\hat{y_{i}}$ is the prediction score of the model.

## 3. Results and Discussion

To verify the effectiveness of the quantitative evaluation system proposed in this paper, we used correlation analysis to analyze the correlation between motor features and physician scale score and the correlation between model prediction score and the physician scale score separately. Pearson’s correlation analysis was performed between the motor features extracted from kinematic data and the scale scores of the rehabilitation physicians. As shown in Figure 5, a heat map was drawn to show the strength of the correlation between each motor feature and the scale scores. The darker the figure color, the stronger the positive correlation, and the lighter the figure color, the stronger the negative correlation. In addition, the matrix figure in Figure 5 represents the correlative coefficient between the motor features and the scale score, and the value ranges from −1 to 1; the larger the absolute value of the number, the stronger the correlation between motor features and the scale score, which means that the correlative coefficient indicates the effectiveness of the motor features selected in the quantitative evaluation system. As shown in Figure 6, the correlation between the trajectory coincidence and the scale scores is the highest at 0.91, followed by two motor features: average trajectory deviation and average velocity. The heat map shows that the correlations between the seven motor feature parameters selected in this study and the physician’s scale score are all greater than 0.5, meaning that each feature had a high correlation with the physician’s scale score, which further shows that the validity of these features in the evaluation of wrist motor function.

The motor feature data sets of 35 subjects were used as the input of the quantitative evaluation model, and the data set was randomly divided into a training set and a test set (the training set was 70% of the data set, and the test set was 30% of the data set). The evaluation model based on the machine learning algorithm was trained and fitted by the training set, the trained evaluation model was tested and inspected by the test set, and each evaluation index and accuracy of the machine learning algorithm model are calculated separately. The model prediction score was obtained by processing and analyzing the motor data in the test set by the trained machine learning algorithm evaluation models. As shown in Figure 7, the prediction score of the four evaluation models based on machine learning algorithms is compared with the doctor’s scale score. The red scatter plot is used to draw the doctor’s scale score, and the blue fitting curve plots the prediction score obtained by quantitative evaluation models. From the comparison chart of the four models, it can be seen that the evaluation performance of the BP neural network model is the best; the fitting curve of the predicted score basically overlaps with the scattering curve of the doctor’s scale score, while the prediction score performance of the other three models is worse than the BP neural network model according to the remaining three graphs in Figure 7.

As shown in Table 3, the indexes of the four evaluation models, including accuracy, mean absolute error, mean square error, and coefficient of determination, were calculated to compare the performance of the four evaluation models and to verify the effectiveness of the evaluation system. Table 3 shows that the accuracy of RF is 90.98%, the accuracy of SVR is 88.50%, the accuracy of KNN is 89.34%, and the accuracy of BPNN is 94.26%. Among the four quantitative evaluation models based on machine learning algorithms selected in this study, the BP neural network has the highest accuracy of 94.26%, the smallest mean square error of 3.6967, the largest coefficient of determination of 0.9284, and the mean absolute error is 1.1393, which is only slightly larger than the mean absolute error of random forest of 1.073. It can be seen that the quantitative evaluation model based on BP neural network has the best performance in evaluating the wrist motor function of stroke patients.

The reason BPNN has a higher accuracy over the others can be attributed to the fact that the BPNN model used multilayer perception as the method for the interconnection of layers of nodes; it has input layers and hidden layers and output layers to establish data processing model [30]. Seven motor features were extracted in this paper as the data of input layers; then, it automatically performed backpropagation and adaptively learnt to achieve better performance. On the other hand, we can take different measures to improve the performance of the remaining three models. The RF model needs to expand the dataset and adjust the parameter values to avoid over-fitting and improve evaluation performance. The SVR model can select different kernel functions to build a better evaluation model. The KNN model could change the ‘K’ value and the method of calculating the distance to obtain high accuracy.

Correlation analysis was performed between the prediction score obtained from the evaluation model and the physician scale score using Pearson’s correlation analysis and Spearman’s correlation analysis, which validated the effectiveness and performance of the quantitative evaluation model and the wrist motor function evaluation system proposed in this paper. Specifically, Pearson’s correlation test and Spearman’s correlation test were used to analyze the correlation between the scale score of rehabilitation physicians and the model prediction score. Statistical analysis was performed using SPSS 22 (IBM Co., Armonk, NY, USA).

Pearson’s correlation analysis between the prediction score of the models and the scale score of the doctor is shown in Figure 8. The black dots represent the comparison between the model prediction score and the doctor’s scale score. The blue line represents the fitted curve between the two, and Pearson’s coefficient of each model is marked in red separately. Pearson’s correlation coefficient of the BP neural network is the highest at 0.964, and the trend performance of the scatter and the fitted curve in the correlation analysis diagram is also better than the other three models.

As shown in Table 4, Pearson’s correlation coefficient between the prediction score of the evaluation model based on the random forest algorithm and the doctors’ scale score is 0.961, the correlation coefficient of the evaluation model based on the support vector machine algorithm is 0.946, and the correlation coefficient of the K nearest neighbor algorithm is 0.953, the correlation coefficient of the evaluation model based on the BP neural network algorithm is 0.964. Moreover, the correlation coefficient between the prediction score of the BP neural network evaluation model and the doctors’ scale score was the highest in Spearman’s correlation test. In addition, the *p*-values of the four evaluation models are all less than 0.05. There is a significant correlation between the prediction score of four evaluation models and the scale score of rehabilitation physicians.

In this paper, a quantitative evaluation system for evaluating wrist motor function in stroke patients was proposed based on the commercial haptic force feedback robot Novint Falcon. Patients completed the experimental task of wrist evaluation by manipulating the force feedback robot, during which they received force feedback from Novint Falcon and visual feedback from the real-time trajectory of the upper computer control software. The kinematic data of the subjects were collected by using the built-in sensors of the haptic device Novint Falcon, as well as the magnitude of the force feedback and the time required to complete the task under different conditions were recorded. The number of peak speed points, average speed, average acceleration, average trajectory deviation, trajectory coincidence, and other motor features was extracted from the kinematic data and used as the input data set for the quantitative evaluation model. In the article, four supervised machine learning algorithms, namely random forest regression, support vector machine regression, K-nearest neighbor, and BP neural network, were selected for building a quantitative evaluation model of wrist motor function. The differences in evaluation performance between the models were compared by calculating the indicators of the models.

Pearson’s correlation between the motor feature parameters and the scale score indicated that the feature parameters characterizing the patient’s wrist motor function obtained by feature extraction have a strong correlation with clinical scale scores. The experimental results show that the accuracy of random forest, support vector machine, K-nearest neighbor, and BP neural network is 90.98%, 88.50%, 89.34%, and 94.26%, respectively, in the prediction score of wrist motor function. It can be seen that the BP neural network has the highest accuracy rate. In the index of mean square error, the BP neural network model has the smallest value of 3.6974, while the support vector machine has the largest mean square error of 5.9344. In the index of mean absolute error, the support vector machine has the largest mean absolute error of 1.4918, and random forest has the smallest of 1.073, while the mean absolute error of BP neural network is only larger than the random forest at 1.1393. In addition, in the index of determination coefficient, the maximum value is the BP neural network of 0.9284, and the minimum value is the support vector machine of 0.8820. The BP neural network algorithm has better evaluation performances for the evaluation of wrist motor function in stroke patients in this study, while the prediction performance of random forest and K nearest neighbor model is similar, and the support vector machine performed slightly worse in terms of evaluation performance. The BP neural network is a multilayer feed-forward neural network with error back propagation [34], and due to the small sample size in this study, it is easier to obtain the expected output value with less error by using the calculation process of BP neural networks in the input layer, hidden layer, and output layer.

Given the consideration of related research, Otten et al. [19] recruited eight subjects to verify the framework of automatic evaluation of upper limb sports injuries established by a machine learning algorithm, and the accuracy of the model was over 90%. Wang et al. [27] recruited 15 stroke patients to participate in the clinical assessment experiment, the results show that classification accuracy could be further enhanced to 96.06% (that is, a multi-modal fusion scheme for comprehensively analyzing the upper-limb motor function’s feasibility). Zhang et al. [31] recruited 21 subjects for the evaluation of upper limb motor function, and the accuracy of the evaluation model was 87.1%. Compared with this paper, the similarity of these studies is the establishment of the evaluation model or scheme based on machine learning algorithms, and it has a greater improvement in accuracy than traditional clinical assessment methods.

The correlation analysis showed that in the correlation test between the model prediction score and the doctor’s scale score, Pearson’s correlation coefficient between the BP neural network model score and the scale score was 0.964, which was greater than the other three models selected in the article. The *p*-values of all four models are less than 0.05, indicating that the prediction score of the four machine learning algorithm models selected in the article is all significantly correlated with the doctor’s scale score. Part of the reason for the high correlation can be attributed to the selection of the doctor’s scale score as the training label during model training. According to clinical rehabilitation physicians, the error of four models in this paper is clinically acceptable, which indicates that the size of the sample in this paper is enough for model establishment. On the other hand, the small size of samples could limit the performance of the evaluation model, and more clinical trials are needed to improve the performance of the evaluation model.

To address the problems of traditional assessment, which is coarsely graded, lacks quantitative analysis, and is time consuming, we proposed a quantitative wrist motor function evaluation system for stroke patients in the article. The clinical feasibility and effectiveness of the system were verified experimentally. Our study has significance in the progress of rehabilitation, which helps reduce clinicians’ workload, grasp the rehabilitation status of stroke patients, and develop individualized rehabilitation programs for them.

The limitation of the article is the small sample size of patients recruited during the experiment, and more clinical experiments are needed to validate the performance of the evaluation model as well as to further improve the prediction evaluation accuracy of the model. On the other hand, more sensors can be used to verify the influence of the compensatory effects of the upper limbs, including the shoulder, during the patient’s performance of the task. Relevant studies have shown that EMG signals have a good correlation with the motor function of the upper limbs of stroke patients [27,35]. The surface EMG sensor detects the subject’s wrist EMG signal as one of the parameters characterizing the patient’s motor function, which can monitor and evaluate the patient more comprehensively. Based on the haptic force feedback robot Novint Falcon, the rehabilitation training of wrist motor function and quantitative evaluation are further combined to develop a remote rehabilitation system or platform that inherits rehabilitation training and assessment functions in one to realize remote rehabilitation training and evaluation functions at home or in the community. This will also be conducive to reducing the workload of rehabilitation doctors while proposing individualized rehabilitation training plans for patients at different stages.

## 4. Conclusions

In this paper, a quantitative evaluation system of wrist motor function in stroke patients based on force feedback and machine learning algorithm is proposed. The main contribution of this work is to establish a quantitative evaluation model of wrist motor function based on a haptic force feedback robot and four machine learning algorithms, which can enable refinement and the quantitative evaluation of wrist motor function in stroke patients. Twenty-five stroke patients and ten healthy volunteers were recruited to participate in the experiment, and rehabilitation doctors were invited to evaluate the subjects’ performance on the clinical scale. The experimental results show that BPNN has the best performance among the four evaluation models, with an accuracy of 94.26%. In addition, there was a significant correlation between the model’s predicted score and the physician’s scale score. The quantitative evaluation system proposed in this article can effectively help clinicians to evaluate the wrist motor function of stroke patients and formulate personalized rehabilitation programs for them. In the future, this study will further conduct more clinical trials to improve the performance of the evaluation model and explore experimental protocols that combine rehabilitation training with clinical evaluation and develop remote home rehabilitation applications for stroke patients.

## Figures and Tables

**Figure 1 sensors-22-03368-f001:**
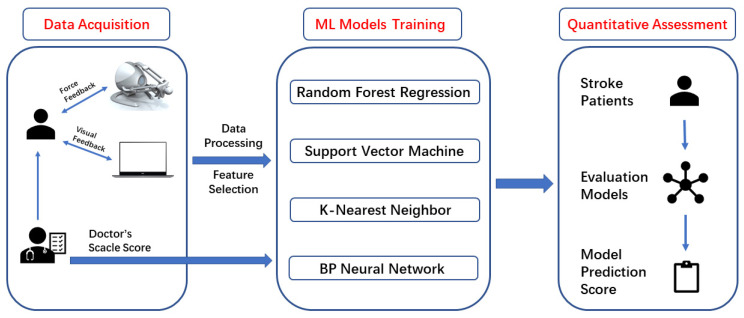
System framework for quantitative evaluation of wrist motor function based on force feedback and machine learning algorithms.

**Figure 2 sensors-22-03368-f002:**
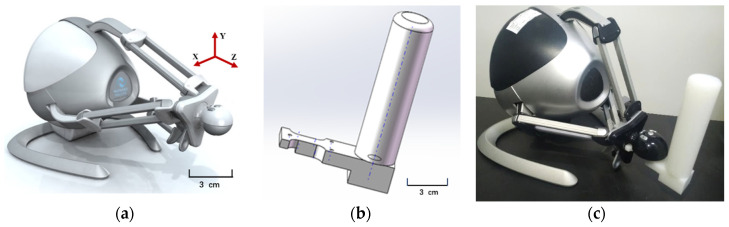
(**a**) Haptic force feedback robot Novint Falcon. (**b**) 3D parts design drawing of the modified handle. (**c**) Effect diagram of the modified handle installed on the Novint Falcon.

**Figure 3 sensors-22-03368-f003:**
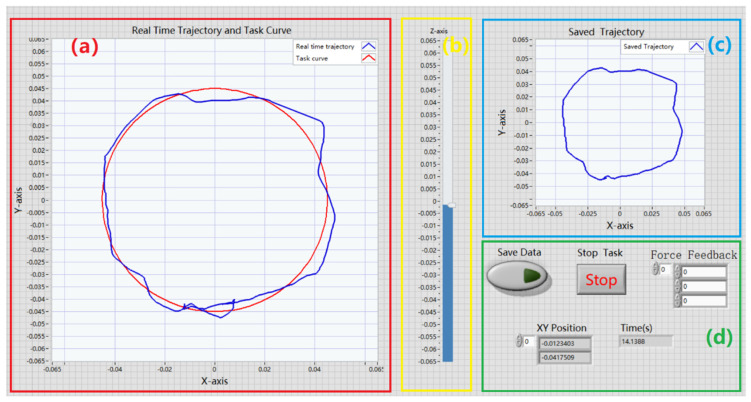
Data acquisition interface of the upper computer control system. (**a**) Real time trajectory and task curve in the experiment. (**b**) Z-axis of the spatial coordinate system. (**c**) Saved trajectory in the experiment. (**d**) the operation buttons and data output part of the upper computer control system.

**Figure 4 sensors-22-03368-f004:**
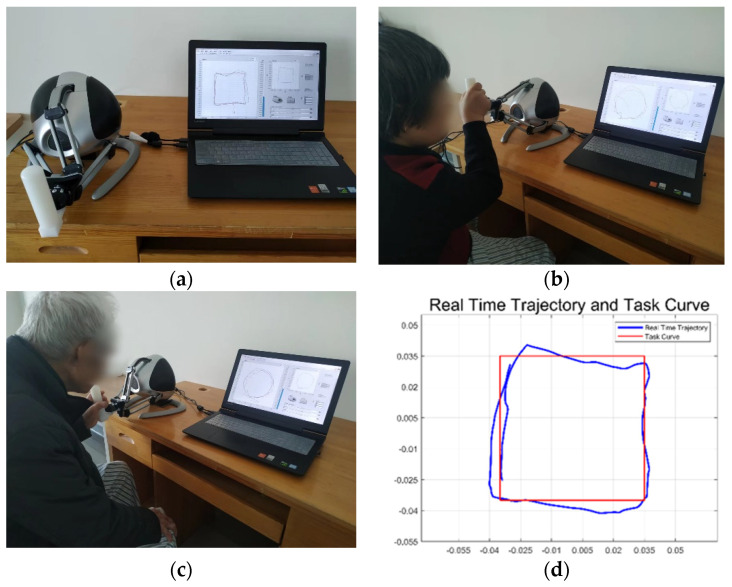
Scene of rehabilitation evaluation experiment and the movement trajectory of the subject. (**a**) The experimental scene of wrist motor function assessment. (**b**) The female subject participated in the wrist assessment experiment by manipulating the Novint Falcon end modification handle with her right hand. (**c**) The male subject participated in the wrist evaluation experiment by manipulating the Novint Falcon end modification handle with his left hand. (**d**) The comparison between the real-time trajectory of the subject and the given graphical curve during the experiment.

**Figure 5 sensors-22-03368-f005:**
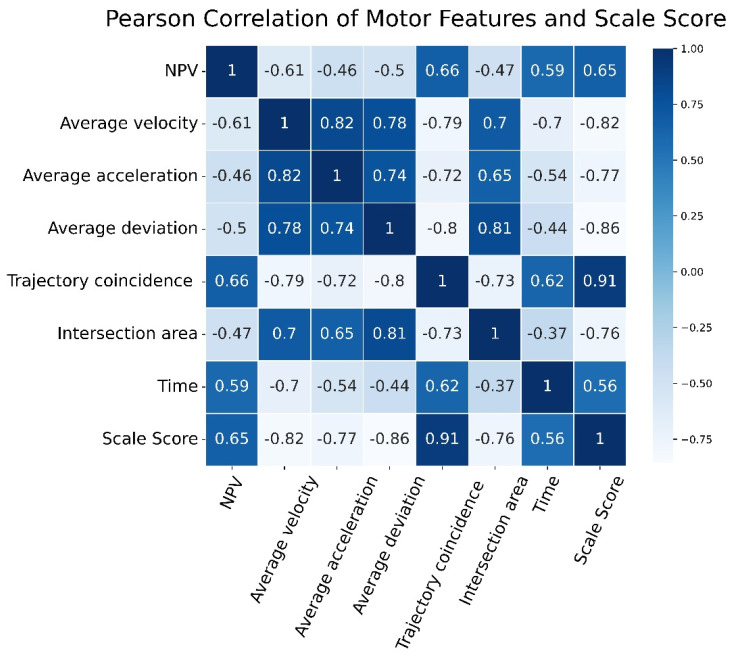
Heat map of Pearson’s correlation analysis between motor features and clinical doctors’ scale score.

**Figure 6 sensors-22-03368-f006:**
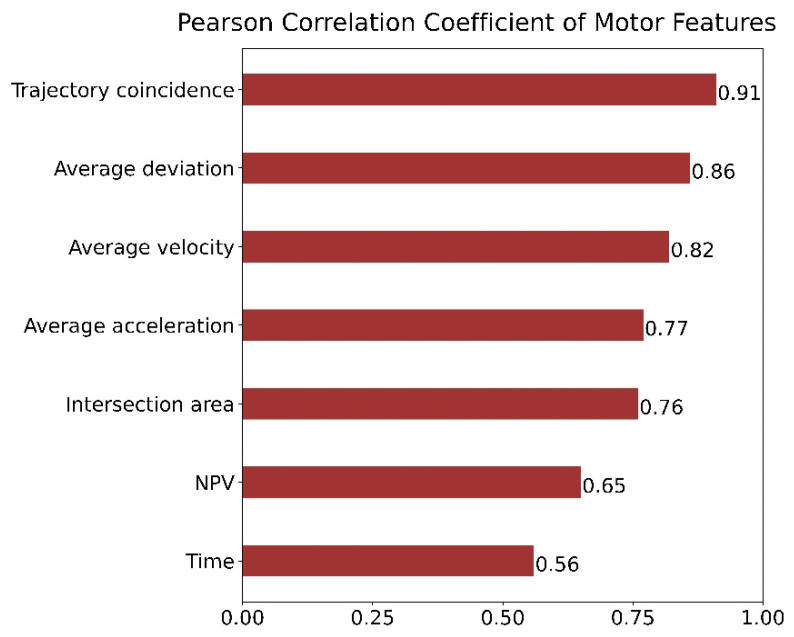
Pearson correlation coefficient of motor features.

**Figure 7 sensors-22-03368-f007:**
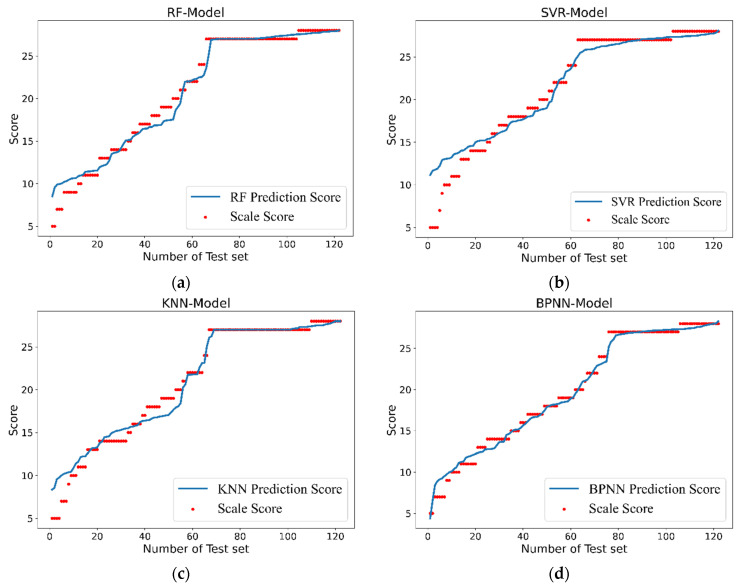
Comparison of the prediction score of the four evaluation models with the doctor’s scale score. (**a**) RF-Model, (**b**) SVR-Model, (**c**) KNN-Model, and (**d**) BPNN-Model.

**Figure 8 sensors-22-03368-f008:**
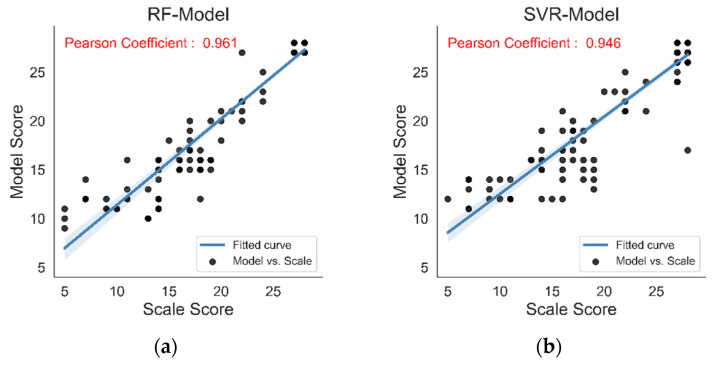
Pearson correlation analysis of the prediction score of four evaluation models and the doctor’s scale score. (**a**) RF-Model, (**b**) SVR-Model, (**c**) KNN-Model, and (**d**) BPNN-Model.

**Table 1 sensors-22-03368-t001:** Demographic information of patients with wrist motor function dysfunction.

Number	Age	Gender	Affected Side	Months after Stroke	Brunnstrom	Scale Score
S1	80	Female	Left	1	II	5
S2	57	Male	Right	1	III	11
S3	46	Male	Right	2	V	19
S4	78	Male	Left	3	IV	17
S5	73	Female	Left	1	III	13
S6	54	Female	Left	4	VI	24
S7	73	Female	Left	3	V	20
S8	69	Male	Right	3	VI	22
S9	79	Female	Left	1	II	7
S10	73	Female	Left	2	III	14
S11	78	Male	Left	1	IV	16
S12	73	Male	Left	3	V	18
S13	38	Male	Right	2	V	21
S14	73	Female	Left	5	IV	16
S15	63	Male	Left	1	III	14
S16	62	Female	Right	3	III	13
S17	56	Male	Right	2	V	19
S18	64	Male	Right	1	IV	17
S19	49	Male	Left	9	VI	22
S20	69	Male	Right	1.5	III	11
S21	76	Male	Right	6	II	9
S22	75	Male	Left	1	IV	14
S23	77	Female	Left	2	IV	15
S24	74	Male	Left	2	III	10
S25	58	Male	Left	6	V	18

**Table 2 sensors-22-03368-t002:** Feature parameters that characterize the motor function.

Feature Parameters	Definition
Number of peaks velocity points NPV	Defined as the number of points on the velocity curve where the instantaneous velocity value is larger than the average velocity.
Average velocity V	Defined as the average of the instantaneous velocity during the subject’s manipulation of the handle movement.
Average acceleration A	Defined as the average of the acceleration during the subject’s manipulation of the handle movement.
Average trajectory deviation D	Defined as the average deviation of the closest distance between the actual trajectory and the given curve.
Trajectory coincidence C	Defined as the ratio of the overlap length between the actual trajectory and the given curve to the actual trajectory length.
Intersected area of trajectory ΔS	Defined as the area formed by the intersection area between the actual trajectory and the given curve.
Task execution time T	Defined as the duration of each task.

**Table 3 sensors-22-03368-t003:** Comparison of evaluation indexes of four evaluation models.

Index	RF	SVR	KNN	BPNN
Accuracy	90.98%	88.50%	89.34%	**94.26%**
MAE	**1.073**	1.4918	1.3524	1.1393
MSE	4.0164	5.9344	4.50	**3.6967**
$R^{2}$	0.9165	0.8820	0.9055	**0.9284**

**Table 4 sensors-22-03368-t004:** Correlation analysis between the model’s prediction score and the doctor’s scale score.

Coefficient	RF	SVR	KNN	BPNN
Spearman	0.929	0.919	0.933	**0.940**
Pearson	0.961	0.946	0.953	**0.964**

## Data Availability

The data used in this study are available from the corresponding authors upon reasonable request. The data are not publicly available because of participant confidentiality.

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
