# Peer review of "Quantitative Evaluation System of Wrist Motor Function for Stroke Patients Based on Force Feedback"

_sensors, 2022, doi:10.3390/s22093368_

Round 1
Reviewer 1 Report
Thank you for submitting your paper. The work done here draws attention to a significant subject sensors used in medical applications. I have found the paper to be interesting. However, several issues need to be addressed properly before the paper is being considered for publication. My comments including major and minor concerns are given below:
- Please consider reviewing the abstract and highlight the novelty, major findings, and conclusions. I suggest reorganizing the abstract, highlighting the novelties introduced. The abstract should contain answers to the following questions:
- What problem was studied and why is it important?
- What methods were used?
- What conclusions can be drawn from the results? (Please provide specific results and not generic ones).
- The abstract must be improved. It should be expanded. Please use numbers or % terms to clearly shows us the results in your experimental work.
- Please consider reporting on studies related to your work from mdpi journals.
- Figure 2 add scale bar or dimensions for the images.
- Line 180 “was used” please use the correct tenses/check grammar in the article.
- Figure 3 better to use (a) and (b) instead of 1 and 2.
- Figure 4 please consider hiding the faces of the participants for ethical reasons.
- Line 350-362 combine into one larger paragraph.
- In Table 1 there are 25 participants, is the size of the sample enough for your models? How can this number affect the accuracy or repeatability of your results? Please discuss and justify.
- Results change to 3. Results and discussion.
- Line 399 “which performs slightly worse” when the author use words such as slightly worse, it is not clear to the readers. It is better to use % terms of numbers to clearly tell us how “slightly worse” it is? Please check this issue elsewhere.
- In Table 3 what could have improve the accuracy of model BPNNN over the other ones? What are the factors and how could the other models be improved? More discussion is needed for this table and correlate your findings with results from past studies similar to your work or closely related to it.
- Combine sections 3 and 4 and call it Results and discussion (recommended).
- Some of the results are merely described and is limited to comparing the experimental observation and describing results. The authors are encouraged to include a more detailed results and discussion section and critically discuss the observations from this investigation with existing literature.
- Conclusion can be expanded or perhaps consider using bullet points (1-2 bullet points) from each of the subsections.
Author Response
Dear Reviewer:
On behalf of my co-authors, I thank you very much for giving us an opportunity to revise our manuscript (sensors-1664886). We appreciate you very much for your careful review and constructive suggestions with regard to our manuscript. These comments are very helpful for us to revise and improve our paper. We have studied the comments carefully and tried our best to revise and improve the manuscript. We have made changes in the manuscript according to the good comments. We appreciate very much your work and hope that the corrections will meet with approval.
A point-by-point response to the comments has been completed, please see the attachment.
Thank you and best regards.
Yours sincerely,
Kangjia Ding

Reviewer 2 Report
-In 'Figure 1', all these models were used for training the model? What was the reason to select these models?
-How was performed the test and validation? which validation method was used?
-What is the accuracy of the proposed model?
-What represent figure 3? What is the role of figure 3 and 4?
-Figure 4a-c should be improved. The device used is not visible.
-In 'Table 1' - What represent the scale score and how was calculated.
R293: The scripts for data processing are missing.
R324: What machine leaning algorithm was used? Can you presents some functions used in this 'machine learning algorithm'?
-The machine learning algorithm is missing.
-What is the role of matrix "figure"? Is the correlative coefficient between scale score and NPV? what means that?
-What represent the figure 8 a-c?
Author Response

(The authors gave the same response as above.)

Round 2
Reviewer 1 Report
The authors have answered all questions from the first round of review and paper can be accepted. Congratulations to the authors!
Author Response
Dear Reviewer:
On behalf of my co-authors, I thank you very much for allowing us to revise our manuscript (sensors-1664886) again. We thank you very much for your nice comments about our manuscript. The comments are constructive for us to revise and improve our paper. we rechecked the grammar and language style of this paper and focused on spelling issues in the article. According to your suggestions for minor spell check, we modified the corresponding problems in the article. We appreciate your work and hope that the corrections will meet with approval.
A response to the comments has been attached, and please see the attachment.
Thank you and best regards.
Yours sincerely,
Kangjia Ding
Reviewer 2 Report
What is the 'other method' used for data processing?
What variable was used for data normalization?
How many neurons were used for training, tests and validation?
What methods were used for validation?
How is scored the rehabilitation? Which algorithm was used? or how do you appreciate the progress in rehabilitation?
Author Response
Dear Reviewer:
On behalf of my co-authors, I thank you very much for allowing us to revise our manuscript (sensors-1664886) again. We appreciate you very much for your careful review and suggestions about our manuscript. These comments are constructive for us to revise and improve our paper. We have studied the words carefully and tried our best to revise and improve the manuscript. We have made changes in the manuscript according to your comments, and the revised portion was marked in red in the marked-up version manuscript. We appreciate your work and hope that the corrections will meet with approval.
A point-by-point response to the comments has been completed, and please see the attachment.
Thank you and best regards.
Yours sincerely,
Kangjia Ding
